# Diagnostic Accuracy of 10/66 Dementia Protocol in Fijian-Indian Elders Living in New Zealand

**DOI:** 10.3390/ijerph18094870

**Published:** 2021-05-03

**Authors:** Adrian Martinez-Ruiz, Rita Krishnamurthi, Ekta Singh Dahiya, Reshmi Rai-Bala, Sanjalin Naicker, Susan Yates, Claudia Rivera Rodriguez, Gary Cheung, Makarena Dudley, Ngaire Kerse, Sarah Cullum

**Affiliations:** 1Department of Psychological Medicine, The University of Auckland, Level 3, Building 507, 22-30 Park Avenue, Grafton, Auckland 1142, New Zealand; a.martinez@auckland.ac.nz (A.M.-R.); deepakreshmi@xtra.co.nz (R.R.-B.); s.nckr@hotmail.com (S.N.); susan.yates@auckland.ac.nz (S.Y.); g.cheung@auckland.ac.nz (G.C.); 2Department of Demographic Epidemiology and Social Determinants, National Institute of Geriatrics, Anillo Perif. 2767, San Jerónimo Lídice, La Magdalena Contreras, Ciudad de México 10200, Mexico; 3National Institute for Stroke and Applied Neurosciences, Auckland University of Technology, 55 Wellesley Street East, Auckland 1010, New Zealand; rita.krishnamurthi@aut.ac.nz (R.K.); ekta.dahiya@aut.ac.nz (E.S.D.); 4Department of Statistics, The University of Auckland, Level 3, Building 303, 38 Princes Street, Auckland 1010, New Zealand; c.rodriguez@auckland.ac.nz; 5School of Psychology, The University of Auckland, Science Centre Level 2, Building 302, 23 Symonds Street, Auckland 1010, New Zealand; m.dudley@auckland.ac.nz; 6School of Population Health, The University of Auckland, Level 3, Building 507, 22-30 Park Avenue, Grafton, Auckland 1142, New Zealand; n.kerse@auckland.ac.nz

**Keywords:** dementia, Alzheimer’s diseases, dementia prevalence, 10/66 dementia protocol, Fijian-Indian community

## Abstract

The 10/66 dementia protocol was developed as a language and culture-fair instrument to estimate the prevalence of dementia in non-English speaking communities. The aim of this study was to validate the 10/66 dementia protocol in elders of Indian ethnicity born in the Fiji Islands (Fijian-Indian) living in New Zealand. To our knowledge, this is the first time a dementia diagnostic tool has been evaluated in the Fijian-Indian population in New Zealand. We translated and adapted the 10/66 dementia protocol for use in in Fijian-Indian people. Individuals (age ≥ 65) who self-identified as Fijian-Indian and had either been assessed for dementia at a local memory service (13 cases, eight controls) or had participated in a concurrent dementia prevalence feasibility study (eight controls) participated. The sensitivity, specificity, positive predictive value, and Youden’s index were obtained by comparing the 10/66 diagnosis and its sub-components against the clinical diagnosis (reference standard). The 10/66 diagnosis had a sensitivity of 92.3% (95% CI 70.3–99.5), specificity of 93.8% (95% CI 75.3–99.6), positive predictive value of 92.3% (95% CI 70.3–99.5), and negative predictive value of 93.8% (95% CI 75.3–99.6). The study results show that the Fijian-Indian 10/66 dementia protocol has adequate discriminatory abilities to diagnose dementia in our sample. This instrument would be suitable for future dementia population-based studies in the Fijian-Indian population living in Aotearoa/New Zealand or the Fiji-Islands.

## 1. Introduction

Dementia is a global public health priority [1]. It is a neurodegenerative disorder that interferes with a person’s ability to live independently by affecting their cognitive abilities and causing behavioural and psychological symptoms [2]. Dementia is one of the most significant causes of disability and dependency among older people. People living with dementia often require support from family members and those around them [2]. The diagnosis of dementia requires a comprehensive assessment including clinical interview, cognitive testing, collateral information from carer or relatives, functional assessment, neuroimaging, and clinical judgment. The individual’s social and cultural background must also be considered in the dementia diagnosis (e.g., primary language) [3].

Aotearoa/New Zealand (NZ) is a bicultural country comprised of Māori and non-Māori. The non-Māori population includes NZ Europeans, Asian, Pacific peoples, and other ethnic minorities. As in many other countries worldwide, the NZ population is ageing; the prevalence of dementia in NZ is estimated to increase from 60,000 in 2015 to 170,000 in 2050 [4]. The increase will be more significant amongst Māori and other ethnic minorities due to the more rapid population ageing in these groups [4]. For instance, the total share of dementia cases in the Māori population is estimated to increase from 5.1% in 2016 to 8.0% in 2038, in Asians from 5.1% to 11.7%, and in Pacific peoples from 2.3% to 3.3%, compared to a decrease in NZ Europeans from 87.5% in 2016 to 77% in 2038 [4].

Fijian-Indians are an ethnic group from the Pacific Islands of Fiji who were originally descendants of a population that migrated from Eastern India to the Fijian Islands between 1879 and 1916. The assimilation of this migrant population into Fiji was followed by the development of an Indo-Fijian culture with unique characteristics and language [5,6]. The Fijian-Hindi dialect is unique to the Fiji-Islands [7]. It is derived from India’s north-eastern and southern languages, primarily Awadhi, Bhojpuri, and Hindustani dialects [8]. It has some similarities to Hindi spoken on the Indian subcontinent, but the two are not interchangeable. Fijian-Hindi has incorporated many linguistic characteristics obtained from the native language of the Fiji-Islands as well as from English. Thus, the Fijian-Hindi language displays a unique variety of idiomatic lexicon and grammatical characteristics from all these languages [8]. Aotearoa/NZ experienced an increase in its Fijian-Indian population in the early years of the twenty-first century, mainly due to political instability in the Fiji Islands [9,10,11]. The 2018 NZ Census estimated that 15,132 Fijian-Indians were living in NZ, the vast majority in the largest city of Auckland, representing less than 1% of the total population [12].

The prevalence of dementia among the Fijian-Indian community is unknown. Their unique genetic make-up [13] and dementia risk factors profile differs from other ethnic groups. Older adults in the Fijian-Indian community have a high prevalence of cardiovascular and metabolic risk factors that might be associated with dementia. For example, a study on the incidence and prevalence of type 2 diabetes mellitus (T2DM) conducted in the Fijian Islands showed that Fijian-Indians have a higher prevalence of T2DM compared to i-Taukei (major Indigenous population of the Fiji Islands also known as Fijian-Melanesians) [14]. Moreover, a similar study using ethnic-specific cut-off points, reported that compared to i-Taukei, Fijian-Indians not only tend to have higher rates of T2DM but also higher rates of obesity [15]. Although the prevalence of T2DM has increased in both Fijian-Indians and i-Taukei, the largest increase occurred in the oldest groups [15]. Fijian-Indian men with T2DM were found to have higher rates of cardiovascular diseases as well as higher mortality rates than the i-Taukei population [16]. Another study conducted using NZ national health records also found a higher prevalence of coronary heart disease among Indian men (including Fijian-Indian) compared to men from other ethnic groups [17]. The greater prevalence of T2DM and cardiovascular diseases may have a greater but yet unknown impact on the burden of dementia in the Fijian-Indian population.

The full impact and burden of dementia can only be understood by estimating the accurate prevalence of dementia. It will help to determine the level of service needs and develop culturally appropriate dementia services, reduce stigma, and increase public awareness. However, the accurate diagnosis of dementia is challenging in populations with varying levels of literacy, sociodemographic settings, and spoken languages. The 10/66 dementia research group recognised the importance of developing a culture and education-fair standardised instrument to diagnose dementia in population-based studies [18,19,20]. To date, the 10/66 dementia protocol has been translated and validated in many different languages, including Spanish, Portuguese, Urdu, Arabic, and Mandarin [20,21,22]. It has a sensitivity and specificity of up to 94% [20]. The aims of this study are (i) to translate and adapt the 10/66 dementia protocol for use in research within the Fijian-Indian community and (ii) to test the diagnostic accuracy of the adapted version.

## 2. Materials and Methods

### 2.1. Index Test: 10/66 Dementia Protocol

Using coefficients derived from the Geriatric Mental State (GMS) [23], the Community Screening Instrument for Dementia (CSI-D) [24], and the delayed recall memory test scores from the Consortium to Establish a Registry of Alzheimer’s instrument (CERAD) [25], the 10/66 dementia research group developed an algorithm to diagnose dementia in population-based studies [20,26]. The 10/66 dementia protocol takes approximately ninety minutes to administer. It has been extensively described elsewhere [20,26], and the three main sections are summarised in Table 1.

#### Translation and Adaptation of the 10/66 Dementia Protocol

We adopted a translation procedure based on the World Health Organisation translation guidelines that demonstrated conceptual equivalence when applied to the adaptation of the Composite International Diagnostic Interview from English into the Malay language [38,39]. It included four steps: (i) Forward translation: the 10/66 dementia protocol in English was translated to Hindi first and then to Fijian-Indian by two bicultural translators with degrees in health/social sciences and fluent in English, Hindi, and Fijian-Indian, (ii) Cross-checking by one bilingual–bicultural clinician with expertise in dementia, (iii) Pre-testing and interviewing four people with and without dementia: feedback was obtained and used to tailor the final instrument, and (iv) The final translated and adapted version was used in the validity study.

### 2.2. Validity Study

#### 2.2.1. Settings and Participants

For this study, the diagnostic accuracy was calculated by comparing the 10/66 dementia protocol results against clinical diagnosis (reference standard). This reference standard has been extensively used in similar 10/66 validity studies [20,21,22].

We recruited the majority of the study participants from the local memory service, and additional controls were recruited from a separate dementia prevalence feasibility study that was running concurrently. We performed a sample size calculation using the method described by Bujang et al. 2016 [40]. In our sample size calculation, we used a prevalence of dementia of 60% based on our previously published data from the memory service [41]. We found a minimum sample size of 52 subjects (including 31 subjects having dementia) were required to achieve a minimum power of 80% for detecting a change in the percentage value of sensitivity of a screening test from 0.70 to 0.90, which was based on a target significance level of 0.05.

#### 2.2.2. Memory Service-Based Participants

The memory service-based participants were recruited from the Counties Manukau Health Memory Service in South Auckland, NZ. The main criteria to access the memory service is a subjective or objective memory complaint perceived by the patient, family members, or a health professional. Individuals are referred to the service from both primary and secondary health care services, and they are assessed in their own homes.

The memory service-based participants were recruited for the study in four steps. Firstly, potential participants were identified at the memory service weekly team meetings. At these meetings, newly referred and previously assessed cases were reviewed by the memory service team. Potential participants were selected for the study if they were 65 years of age or over, self-identified as Fijian-Indian, and met the other eligibility criteria for the study described below. A member of the clinical team approached the patients to give preliminary details of the study and request consent to pass on their contact details to the research team. Following this, a research assistant contacted potential participants by phone, explained the study’s aims, and invited them to participate. The study information sheet and informed consent were sent to the interested participants by post. Finally, potential participants were contacted again by phone to answer any questions about the study. If they were still interested in participating, and had an informant willing to participate, an appointment for consent procedures and the research interview was scheduled at a convenient time and place (usually the person’s own home).

The clinical diagnosis (reference standard) was made by the memory service team at the weekly multidisciplinary team meeting using standard dementia clinical criteria, including Diagnostic and Statistical Manual of Mental Disorders 4th edition (DSM-IV) dementia criteria [28], National Institute of Neurological and Communicative Diseases and Stroke/Alzheimer’s Disease and Related Disorders Association (NINCDS-ADRDA) criteria for Alzheimer’s disease dementia [42], National Institute of Neurological Disorders and Stroke and Association Internationale pour la Recherché et l’Enseignement en Neurosciences (NINCDS-AIREN) criteria for vascular dementia [43], criteria for Lewy Body dementia [44], and Lund criteria for frontotemporal dementia [45]. The participants were categorised as having “clinical dementia diagnosis” or “no clinical dementia diagnosis” (the latter category included people with mild cognitive impairment (MCI)).

##### Eligibility Criteria for Memory Service-Based Dementia Cases

Included cases were those who self-identified as Fijian-Indian, were 65 years or older, had been assessed by the memory service team within six months prior to starting the study, and were diagnosed with dementia using standard clinical protocols and investigations as described above. Participants that were unable to complete the interview due to a physical or sensory impairment (not due to dementia), or who did not have an informant were excluded from the study.

##### Eligibility Criteria for Memory Service-Based Controls

Included controls were those who self-identified as Fijian-Indian, were 65 years or older, and were assessed as not having dementia by the memory clinic team in the six months prior to starting the study. All controls were assessed using the same criteria as described above. The same exclusion criteria as in service-based dementia cases also applied to memory service-based controls. Following the criteria for the quality assessment of diagnostic accuracy studies (QUADAS) [46], we decided to not exclude controls with MCI, as excluding them from the memory-service control group will increase the risk of “spectrum bias”, causing spuriously accurate results.

#### 2.2.3. Community-Based Participants

The community-based study participants were recruited from a sample identified from a concurrent study aimed at assessing the feasibility of conducting a dementia prevalence study in two areas of the community served by Counties Manukau Health. The community-based controls did not have a full memory service assessment; instead, they were included as controls in the study if they scored ≥27 on the Rowland Universal Dementia Assessment Scale (RUDAS) [47] and 1–3 in the Informant Questionnaire on Cognitive Decline (IQCODE) [48,49,50], which effectively excludes any potential cases of mild dementia. The RUDAS is a cognitive instrument composed of six sections evaluating the following cognitive domains: visuospatial orientation (5 points), praxis (2 points), visuospatial construction (3 points), judgment (4 points), memory recall (8 points), and language (8 points). A maximum total score of 30 can be obtained by a participant. The cut-off for dementia reported in the NZ population is <23 [51]. The IQCODE short version is a 16-item questionnaire used to assess cognitive decline or improvement in everyday functioning over the last ten years [49]. It is completed by a person who knows the patient/participant well enough to give the most accurate responses. The answers are ranked on a 5-point scale (1 = much improved, 2 = a bit improved, 3 = not much change, 4 = a bit worse, 5 = much worse). The final IQCODE score is calculated by adding up the score for each question (1–5) and dividing by the number of questions (16-questions). A total score from 1–5 can be calculated, and an average score of 3.31 or above is indicative of cognitive impairment [52].

Both instruments have demonstrated adequate sensitivity and specificity for dementia in a number of diagnostic test accuracy studies (Table 2) [47,48,49,50,51,52]. For example, a systematic review of eight studies that used DSM-IV criteria as a reference standard [53] found that the RUDAS had an overall sensitivity of 77.2% (95% CI 67.4–84.5) and a specificity of 85.9% (95% CI 74.8–92.6). A Cochrane systematic review of diagnostic test accuracy of the IQCODE [54] included 2745 participants from thirteen studies, and it reported that the pooled sensitivity and specificity was 91% (95% CI 0.86 to 0.94) and 66% (95% CI 0.56 to 0.75) at a cut-point of 3.3, and in general secondary care settings, the sensitivity and specificity of the instrument were higher (95% and 81%, respectively) when compared to specialised memory settings. A study carried out in the Arabic-speaking population combined the RUDAS and IQCODE using a “weighted sum” method [55] and found that combining both instruments resulted in higher predictive validity. Although both instruments have demonstrated adequate clinimetric characteristics, we used considerably higher than normal cut-off points to reduce the possibility of misclassification bias in community-based controls.

##### Eligibility Criteria for Community-Based Controls

We included people living in private residences in the selected areas, who self-identified as Fijian-Indian and were 65 years or older. Participants who were unable to complete the interview due to a physical or sensory impairment, who did not have an informant, and/or were living in a nursing home or retirement village were excluded.

To respect cultural protocols, all phone and face-to-face contacts with potential participants, both memory service-based and community-based, were made by the research team members and conducted in the participant’s preferred language, either English or Fijian-Indian. The study was approved by the New Zealand Northern A Health and Disability Ethics Committee—New Zealand Government/Ministry of Health (Ref: 17NTA234).

#### 2.2.4. Participants Unable to Give Informed Consent

The NZ Code of Health and Disability Services Consumers’ Rights (right number seven) was followed regarding participants who were not able to give fully informed consent [64]. Thus, we sought written confirmation from their caregiver/ next of kin that they agreed to their relative participating in the study and that it would fit with the wishes of their relative. However, if at any time during the interview the participant indicated they did not want to continue or became distressed, the interview was discontinued.

#### 2.2.5. Informants

An informant was defined as a person who knows the main participant well. All memory service-based and community-based participants included in the study had an informant. The informant was usually the primary caregiver, a family member, or someone else in charge of the participant’s care. All informants gave informed consent to participate in the study. The consent form was explained and signed separately to those of the participants.

#### 2.2.6. Blinding

Memory service-based participant interviews: the 10/66 interviewing process was performed independently of the clinical assessment of cases and controls; therefore, interviewers using the 10/66 were blind to participants’ status.

Community-based participant interviews: To minimise unblinding the assessors, the two interviewers initially applied the 10/66 dementia protocol, one of them interviewing the participant and the other one interviewing the informant. Once both interviews were concluded, the interviewers switched roles to apply the reference-standard cognitive testing (RUDAS and IQCODE). Thus, the interviewer who previously interviewed the participant with the 10/66 protocol questionnaire would then apply the reference-standard and vice versa.

#### 2.2.7. Interviewers

This study was conducted by bilingual (Fijian-Indian and English) and bicultural interviewers self-identifying as Fijian-Indian. The interviewers attended four training sessions on the 10/66 dementia protocol. Each session lasted four hours, and all the sections of the instrument were thoroughly covered, including the consent process and protocols to handle unexpected situations. A separate session lasting one and a half hours on how to score the RUDAS and IQCODE was also conducted. Additionally, the first three study interviews were carried out under the trainers’ supervision, and specific feedback regarding the interview process was provided at the end of the interview. This assured that the 10/66 dementia protocol was correctly administered across all participants, and it was an opportunity to clarify any questions raised during the interview process.

#### 2.2.8. Interviewing Process

Once written consent was obtained from the participant and the informant, the interview was carried out following Fijian-Indian cultural protocols. For example, in the Fijian-Indian culture, it is essential to build trust and rapport before conducting the interviews; therefore, before the interview took place, the interviewers shared information about their family backgrounds and their place of origin in Fiji. Furthermore, to show respect, they always referred to the participant and their family members with formal and appropriate Fijian-Indian language with words such as Aap/You and Ji, which are akin to Sir and Madam, respectively. At the end of the session, a New Zealand Dollar $100 voucher was given to the participants and their families as a gift of appreciation for their time.

### 2.3. Data Analysis

#### 2.3.1. Dementia 10/66 Diagnosis

The dementia diagnosis was obtained using the 10/66 dementia diagnostic algorithm. The algorithm organises the final results as “10/66 dementia” or “10/66 no dementia”, depending on the score obtained from the logistic regression equation developed in the 10/66 international pilot study [20]. The equation is based on the predicted probability for DSM-IV dementia syndrome [26], and it uses coefficients obtained from the analysis of the following three sub-components [20]:

(i) The Community Screening Interview for Dementia (CSI-D) [24] composed of the participant section (32 item cognitive test) and the informant section (26 item questionnaires about participant’s cognitive state and functional level). Three summary scores were obtained using the CSI-D questionnaires: (1) The global cognitive score (COGSCORE) from the participant’s questionnaire, “representing item-weighted total score from the participant cognitive test” [20]; (2) The informant’s score (RELSCORE) “representing an unweighted total score from the informant interview” [20]; and (3) The discriminant function weighted score (DFSCORE), which was calculated by combining COGSCORE and RELSCORE [18,20]. The algorithm calculates all three scores but uses only COGSCORE and RELSCORE to allocate 10/66 dementia cases or 10/66 no dementia cases. There are previously established COGSCORE cut-off points for: no dementia (COGSCORE > 29.5), possible dementia (COGSCORE > 28.5 but ≤29.5), and probable dementia (COGSCORE ≤ 28.5), and for DFSCORE: no dementia (DFSCORE < 0.120), possible dementia (DFSCORE ≥ 0.120 but <0.184), probable dementia (DFSCORE ≥ 0.184) [20,65].

(ii) The Consortium to Establish a Registry for Alzheimer’s Disease (CERAD) 10-word list delayed recall [25]. The word list memory test is conducted in two phases. In phase one, the interviewer reads out the 10-word list, and participants have to recall as many words as they remember. This procedure is repeated three times. After 5 min, the interviewer asks the participant to recall the words they remember from the list. The first phase has a possible learning score of up to 30 and the second phase or delayed recall of up to 10 points. The 10/66 algorithm uses coefficients from the delayed recall score.

(iii) The Geriatric Mental State version B3 (GMS/AGECAT) [23] is a clinical interview that, when analysed and processed by the 10/66 algorithm, organises the symptoms into four categories/clusters according to the ICD 10 [27] and DSM-IV [28] clinical criteria. The four categories or clusters are schizophrenia and related psychosis, anxiety neuroses, neurotic and psychotic depression, and organic brain syndrome/dementia [23].

#### 2.3.2. Statistical Analysis

The predictive analytic software version 25 (SPSS, Chicago, IL, USA) was used for data analysis. Descriptive frequency distributions and mean values were used to describe demographic data. The chi-square test and student’s t-test were used to compare categorical and continuous variables among cases and controls. The sensitivity, specificity, positive predictive value (PPV), negative predictive value (NPV), positive likelihood ratio (PLR), negative likelihood ratio (NLR), and Youden’s index (as a summary measure for sensitivity and specificity) were calculated comparing the 10/66 dementia protocol primary outcomes (“10/66 dementia” or “no 10/66 dementia”) against the results of the reference standard diagnosis (“clinical dementia diagnosis” or “no clinical dementia diagnosis”). Since the sub-components had been adapted and translated specifically for the Fijian-Indian culture, the psychometric properties for the 10/66 dementia protocol sub-components were also calculated: CSI-D (using previously established cut-off points) and CERAD delayed recall. The CERAD delayed recall test cut-off point was calculated for this sample using the Area Under the Receiver-Operator Characteristics Curve (AUROC) analysis; a cut-off point of three showed the best sensitivity and specificity for our sample. The AUROC analysis was used to predict the overall performance of the 10/66 protocol sub-components (with 95% confidence intervals). The AUROC analysis used the full range of CSI-D (COGSCORE and RELSCORE), and CERAD delayed recall ordinal raw scores.

## 3. Results

### 3.1. Translation and Adaptation

The following changes were made to the CSI-D participant version: one item for the naming of body parts (“knuckles” was changed to “fist” since there is no common name for “knuckles” in Fijian-Indian), and one item for attention and language (the phrase “no ifs, ands, or buts” was changed for “neither this nor that”). The general knowledge question (the name of the mayor/village head) was adapted according to how long the participant had lived in NZ: if they had lived in NZ for less than 12 months, we asked mayor/village head of Fiji; if they had lived in NZ for more than 12 months, we asked for the prime minister of NZ. Lastly, the item assessing long-term memory was changed from “What is the name of the civil rights leader who was assassinated in Memphis in 1968?” to “Who was the person responsible for executing a military coup on 14 May 1987 in Fiji?” No modifications were made to the translated versions of the GMS, the CERAD l0 word list test, and the CSI-D informant questionnaire.

### 3.2. Demographic Characteristics

Twenty-nine participants were recruited for this study, thirteen participants with a clinical dementia diagnosis, and sixteen with no clinical dementia diagnosis. One of the memory-service based controls had a diagnosis of MCI and was included in the “no dementia” group as per protocol. Two cases of clinical dementia diagnosis were identified in the community-based feasibility study: both were identified patients who had previously attended the memory service and therefore met the eligibility criteria for memory service-based cases. All participants and informants completed the 10/66 dementia protocol (Figure 1). The average age of the participants was 75.0 (SD ± 6.5) years old, and the majority were male (*n* = 20, 69.0%) and married (*n* = 22, 75.9%). Regarding education level, seven of the 16 people in the no clinical dementia diagnosis group had completed primary education (43.8%), compared to four of the 13 people in the clinical dementia diagnosis group (30.8%). The average age of informants was 60.9 (SD ± 18.7) years old, the majority were female (*n* = 20, 69.0%), and most were spouses/partners of the participants (*n* = 22, 75.9%). There were no statistically significant differences in the sociodemographic characteristics between the participants in the clinical dementia diagnosis and no clinical dementia diagnosis groups, except for age; see Table 3.

### 3.3. Diagnostic Test Accuracy

#### 3.3.1. 10/66 Dementia Diagnosis

The sensitivity and specificity of the 10/66 dementia protocol were 92.3% (95% CI 70.3–99.5) and 93.8% (95% CI 75.3–99.6) respectively, a PPV value of 92.3%, NPV of 93.8%, PLR 14.7, NLR of 0.08, and a Youden’s index of 0.86.

#### 3.3.2. 10/66 Dementia Protocol Sub-Components Scores

The CSI-D and CERAD delayed recall test raw scores showed statistically significant differences in all its sub-components between the two groups (Table 4). The CERAD delayed recall test cut-off point was calculated, and a cut-off point of three showed the best sensitivity for our sample. The sensitivity, specificity, PPV, NPV, PLR, NLR, Youden’s index, and AUROC analysis results for the 10/66 protocol sub-scores are shown in Table 5.

## 4. Discussion

The overall sensitivity and specificity of the 10/66 dementia protocol in our study was 92.3% (95% CI 70.3–99.5) and 93.8% (95% CI 75.3–99.6) respectively, a positive predictive value of 92.3% (95% CI 70.3–99.5), negative predictive value of 93.8% (95% CI 75.3–99.6), positive likelihood ratio of 14.7, negative likelihood ratio of 0.08, and Youden’s index of 0.86. Our study showed that the translated and adapted Fijian-Indian version of the 10/66 dementia protocol has adequate sensitivity and specificity. Thus, it is suitable to be used in future dementia population-based studies in NZ, and it demonstrated excellent discriminatory abilities for dementia diagnosis in adults over 65. To our knowledge, this is the first validity study of the 10/66 dementia instrument focusing on Fijian-Indians in NZ and elsewhere.

We used the clinical diagnosis as a reference standard for memory service-based participants, which is accepted and used as the reference standard in clinical dementia studies worldwide [20,21,22]. Its validity has been previously demonstrated in multiple studies. For example, a study on the accuracy of clinical dementia diagnosis, conducted by Lopez et al., compared medical records of subjects with clinical diagnosis of Alzheimer’s diseases (between 1983 and 2000) against their neuropathological diagnosis [66]. Their study showed an overall clinical sensitivity and specificity of up to 97% and 80%, respectively. However, they also found that the sensitivity and specificity varied when analysing different periods of time, being as high as 98% and 88%, respectively [64]. However, a study at the National Institute on Aging Alzheimer’s Disease Centres found that the sensitivity and specificity of the clinical diagnosis may increase or decrease depending on how permissive or restrictive the clinical criteria is [67]. We followed strict clinical criteria in order to reduce the possibility of allocating people without dementia to the clinical dementia diagnosis group.

In regard to the overall sensitivity and specificity of the 10/66 dementia protocol, our results are similar to those found in other studies (Table 6). The sensitivity of the CSI-D scores (COGSCORE and DFSCORE) were both above 90%, and the specificity of the DFSCORE was 93.8% and the COGSCORE of 81.3%. These results replicated those found in other validity studies (Table 6). The sensitivity of the CERAD delayed recall test was 76.9% with a specificity of 85.7%. Other studies have reported similar cut-off points; a study conducted in Brazil showed that the ideal cut-off point in their population was one or three, depending on the literacy level of the subjects [68]. A study conducted in Finland suggested a cut-off for the CERAD delayed recall of less than five in highly educated participants [69].

Some limitations need to be acknowledged. The main limitation is that our study sample was small compared to other studies, and therefore, there will be more uncertainty about the results. We designed our study according to the requirements for minimum sample size (*n* = 52) [40], but we were only able to include full data for 16 controls and 13 dementia cases. This was due to the size of the underlying target population in Counties Manukau and by association the numbers of Fijian-Indian patients attending the Counties Manukau memory service, plus a lower than expected response rate in the community setting due to fears around the COVID-19 pandemic. The proportion of the Fijian-Indian population represents less than 1% of the total population in NZ [12] and even less for Fijian-Indian people aged 65 or over. We believe that even though our study sample is small, it is still essential to have some information about validated dementia tools in the Fijian-Indian community living in NZ. One possible solution to further examine the diagnostic accuracy of the 10/66 dementia protocol is to embed the evaluation within a population-based prevalence study. The latter approach has been used before in other 10/66 dementia prevalence studies [70]. Despite this, our findings did replicate those reported by other studies (as described in Table 6), which gives us some degree of confidence that the 10/66 dementia protocol is a valid instrument to measure the prevalence of dementia in our population. In addition, due to the sample size, our sample did not include other control groups. For example, including control groups with a clinical diagnosis of depression or mild cognitive impairment will help to clarify the 10/66 dementia protocol performance against other diagnoses and levels of dementia severity in our population. Although the 10/66 algorithm can differentiate cognitive impairment due to dementia and cognitive impairment due to other conditions (including depression and MCI), this was not explored in our small sample. Similarly, even though the algorithm results are adjusted by years of education, a larger sample size including participants with a higher and lower level of education would have been preferable. Another limitation of the study is that even though we tried to keep interviewers as blind as possible to avoid misclassification bias, this was especially difficult in cases with moderate to severe dementia. Finally, the instrument’s best performance is in a community setting, but our sample was mixed, including people recruited from the memory service and from community settings. Recruiting from a community-based setting would be ideal for assessing the 10/66 dementia protocol properties in the Fijian-Indian population, but this was beyond our budget.

## 5. Conclusions

Although our sample size was relatively small, we have demonstrated that the translated and adapted Fijian-Indian version of the 10/66 dementia protocol has adequate sensitivity (92.3%, 95% CI 70.3–99.5) and specificity (93.8%, 95% CI 75.3–99.6), PPV (92.3%), NPV (93.8%), and a Youden’s index of 0.86. We are confident that the Fijian-Indian version of the 10/66 dementia protocol would have similar utility as other versions of 10/66 dementia protocols [20,21,22]. To date, the actual extent and impact of dementia in the Fijian-Indian population is unknown. The validated Fijian-Indian 10/66 dementia assessment protocol can be used to conduct a future dementia prevalence study, which in turn, will help to (1) measure the true extent of dementia in the Fijian-Indian population living in NZ; (2) examine its unique risk factor profile; (3) measure caregiver burden; and (4) determine the economic impact of dementia on Fijian-Indian families. This information will be essential to inform the development of culturally appropriate strategies to reduce the impact of dementia and specific policies to raise public awareness about dementia and its prevention in the Fijian-Indian community.

## Figures and Tables

**Figure 1 ijerph-18-04870-f001:**
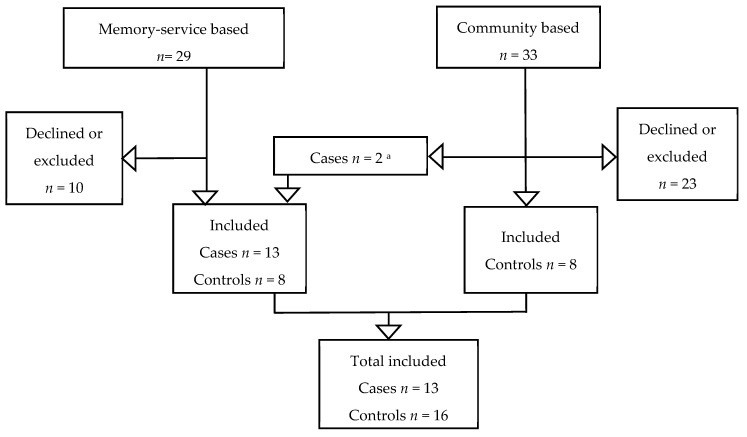
Recruitment flow-chart. ^a^ Two cases were identified in the community-based feasibility study: both were identified patients who had previously attended the memory service and therefore met the eligibility criteria for memory service-based cases.

**Table 1 ijerph-18-04870-t001:** Sections of the 10/66 dementia protocol.

Questionnaire	Section	Instruments Used
1. Participant	1.1 Clinical interview	GMS B3 [23] generates hierarchically organised ICD10 [27] and DSM-IV [28] diagnoses.
	1.2 Cognitive test battery	CSI-D participant version [24]; CERAD word list memory test (immediate and delayed recall) [25]; CERAD verbal fluency test [25]; Neurological examination—Palm-fist-hand test from the Luria battery of frontal lobe tasks [29].
	1.3 Sociodemographic status	Sociodemographic and risk factors questionnaire (participant version) [18,20].
2. Informant	2.1 Informant interview	Brief informant history from the CSI-D [24]; Client Service Receipt Inventory or CSRI [30]; Self-reported questionnaire [31,32]; The Zarit Burden Interview [33,34,35]; History and Aetiology Schedule [36]; Neuropsychiatric Inventory Questionnaire or NPI-Q [37].
	2.2 Sociodemographic status	Sociodemographic and risk factors questionnaire (proxy version) * [18,20].
3. Household	3.1 Head of household questionnaire	Questions about house and family income [18,20].

Abbreviations—ICD: International Classification of Diseases, DSM: Diagnostic and Statistical Manual of Mental Disorders, GMS: Geriatric Mental State; CERAD: Consortium to Establish a Registry for Alzheimer’s Disease; CSI-D: Community Screening Interview for Dementia. NPI-Q: Neuropsychiatric Inventory–Questionnaire. * Proxy version was used if the main participant were unable to complete the participant version of Sociodemographic and risk factors questionnaire.

**Table 2 ijerph-18-04870-t002:** RUDAS and IQCODE sensitivity and specificity in samples from different populations.

Author, Year	Storey, 2004 [47]	Rowland, 2006 [56]	Ozel-Kizil, 2010 [57]	Goncalves, 2011 ^b^ [58]	Sanchez, 2013 [59]	Nielsen, 2013 [60]	Lourenço, 2014 [61]	Cheung, 2015 [51]	Nielsen, 2016 ^b^ [55]	Goudsmit, 2020 ^b^ [62]
Place, ethnic background	Australia Multiple	Australia, Multiple	Turkey, Turkish	Australia, Multiple	Brazil, Brazilian	Denmark, Multiple	Brazil, Brazilian	New Zealand, Multiple	Lebanon, Arabic	The Netherlands, Dutch
Reference standard	DSM-IV criteria	DSM-IV criteria	DSM-IV-TR criteria	DSM-IV-TR criteria	DSM-IV and clinical diagnosis	DSM-IV-TR criteria	DSM-IV criteria	Clinical diagnosis in conjunction with MMSE	DSM-IV criteria	DSM-IV-TR criteria
Total sample size	90	111	220	204	169	142	406	84	225	109
RUDAS										
Cut-off point	<23	<23		<21		<24		<23	<23	<23
Sensitivity	89%	81%		66%		69%		78.40%	82%	80%
Specificity	98%	95.80%		90%		80%		85.10%	84%	59%
IQCODE										
Cut-off point			>3.4	>4.1	>3.51		>3.26		>3.34	>3.7
Sensitivity			82%	72%	83.30%		89%		92%	80%
Specificity			70%	67%	80.70%		72%		96%	74%
Combined ^a^										
Sensitivity									86%	
Specificity									97%	

RUDAS = Rowland Universal Dementia Assessment Scale, IQCODE = Informant Questionnaire on Cognitive Decline, DSM: Diagnostic and Statistical Manual of Mental Disorders, TR= Text Revision, MMSE: Mini-Mental State Examination [63] ^a^ Weighted combination method of the RUDAS and the IQCODE, ^b^ IQCODE short version (16-items).

**Table 3 ijerph-18-04870-t003:** Sociodemographic characteristics of participants and informants by reference standard clinical diagnosis.

Variable	No Clinical Dementia Diagnosis *n* = 16 (%)	Clinical Dementia Diagnosis *n* = 13 (%)	*p*-Value
Mean age (SD)	72.6 (SD ± 5.0)	77.8 (SD ± 7.3)	0.03
Sex (f)	5 (31.3)	4 (30.8)	0.64 ^a^
Marital Status			
Married/cohabitating	13 (81.2)	9 (69.2)	0.37 ^a^
Other	3 (18.8)	4 (30.8)	
Education level			
None	1 (6.2)	0 (0)	0.38
Primary not completed	3 (18.8)	6 (46.2)	
Primary completed	7 (43.8)	4 (30.8)	
Secondary or above	5 (31.2)	3 (23.0)	
Informant mean age (SD)	62.7 (SD ± 16.5)	58.7 (SD ± 21.6)	0.57
Informant sex (f)	10 (62.5)	10 (76.9)	0.33 ^a^
Informant relationship with participant			
Spouse/partner	12 (75)	10 (76.9)	0.62 ^a^
Other	4 (25)	3 (23.1)	

SD = Standard Deviationa, ^a^ Fisher exact test.

**Table 4 ijerph-18-04870-t004:** Analysis of the CSI-D and CERAD delayed recall test raw scores by clinical diagnosis.

	$\mathrm{No}\;\mathrm{Clinical}\;\mathrm{Dementia}\;\mathrm{Diagnosis}\;\bar{x}$ (±SD)	$\mathrm{Clinical}\;\mathrm{Dementia}\;\mathrm{Diagnosis}\;\bar{x}\;(\pm \mathrm{SD})$	*p* Value
CSI-D COGSCORE	30.0 (±4.0)	21.0 (±9.7)	0.002
CSI-D DFSCORE	0.17 (±0.18)	0.64 (±0.35)	0
CERAD delayed recall	5.19 (±2.3)	0.92 (±1.3)	0

CSI-D = Community Screening Instrument for Dementia, CERAD = The Consortium to Establish a Registry for Alzheimer’s Disease.

**Table 5 ijerph-18-04870-t005:** Psychometric properties of 10/66 dementia protocol and its sub-components.

	Sensitivity % (95% CI)	Specificity % (95% CI)	PPV % (95% CI)	NPV % (95% CI)	LR+ (95% CI)	LR− (95% CI)	Youden’s Index	AUROC (95% CI)
10/66 global assessment	92.3	93.8	92.3	93.8	14.7	0.08	0.86	
	(70.3–99.5)	(75.3–99.6)	(70.3–99.5)	(75.3–99.6)	(2.2–99.1)	(0.01–0.54)		
Sub-assessments								
CSI-D COGSCORE	92.3	81.3	80	92.9	4.9	0.09	0.73	0.89
	(70.3–99.5)	(58.3–95.0)	(56.0–94.6)	(72.1–99.6)	(1.7–13.8)	(0.01–0.63)		(0.77–0.99)
CSI-D DFSCORE	100	93.8	92.9	100	16	0	0.93	0.95
		(75.3–99.6)	(72.1–99.6)		(2.3–106.7)			(0.85–0.99)
CERAD	76.9	87.5	83.3	82.4	6.1	0.26	0.64	0.93
delayed recall	(50.5–93.7)	(66.2–97.8)	(56.9–97.0)	(60.4–95.3)	(1.6–23.2)	(0.09–0.72)		(0.83–0.99)

CI = confidence interval, PPV = positive predictive value, NPV = negative predictive value, LR+ = positive likelihood ratio, LR− = negative likelihood ratio, AUROC = Area Under the Receiver-Operator Characteristics Curve, CSI-D = Community Screening Instrument for Dementia, CERAD = The Consortium to Establish a Registry for Alzheimer’s Disease.

**Table 6 ijerph-18-04870-t006:** Comparison of sensitivity and specificity of the 10/66 dementia protocol and its sub-components in samples from different populations.

Author, Year	Present Study	Khan, 2020 [22]	Phung, 2015 [21]	Subramaniam, 2015 [70]	Nozari, 2009 ^a^ [71]	Prince, 2008 [26]	Prince, 2003 [20]
Population	New Zealand	Pakistan	Lebanon	Singapore	Iran	Cuba	India, China, Taiwan, Nigeria, Latin America
Language	Fijian Indian	Urdu	Arabic	English, Chinese, Malay, and Tamil	Farsi	Spanish	Translations into all local languages
Reference standard	Clinical diagnosis	Clinical diagnosis	Clinical diagnosis	Clinical	Clinical diagnosis	Clinical diagnosis	Clinical
				diagnosis			diagnosis
Sample Size (*n*)	29	257	244	2421	120	1887	2885
10/66 global assessment							
Sensitivity	92.30%	70.30%	92%	95.60%	98.30%	93.20%	94%
Specificity	93.80%	91.70%	95.10%	81.80%	98.30%	96.80%	94% ^b^
CSI-D COGSCORE							
Sensitivity	92.30%	86.70%	98%		98.30%		92%
Specificity	81.30%	72.10%	49.30%		81.70%		
CSI-D DFSCORE							
Sensitivity	100%	71.10%	92%		96.70%		95%
Specificity	93.80%	96.10%	90.30%		96.70%		
CERAD delayed recall							
Sensitivity	76.90%	85.90%	91%		93.3% ^b^		
Specificity	87.50%	62.20%	67.40%		90.0% ^b^		

CSI-D = Community Screening Instrument for Dementia, CERAD = The Consortium to Establish a Registry for Alzheimer’s Disease. ^a^ Published as a letter to the editor; ^b^ In people with low education.

## Data Availability

Data will be available upon reasonable request to the corresponding author.

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
