# Peer review of "Diagnostic Accuracy of 10/66 Dementia Protocol in Fijian-Indian Elders Living in New Zealand"

_ijerph, 2021, doi:10.3390/ijerph18094870_

Round 1

Reviewer 1 Report

Comments

  • Authors should further clarify and elaborate novelty in their abstract.
  • Conclusion is too short. Add more explanation.
  • What are the practical implications of the present study? Add them in the conclusion section?

Reviewer 2 Report

The design of this study is in my opinion basically flawed as a diagnostic accuracy study should not be based on only 29 participants.

The reported SN of 92.3 for the 10/66 instrument means (Table 5) that 12 of 13 cases were correctly classified and only one case was misclassified. Similarly, the reported SP of 93.8 means that 15 of 16 controls were correctly classified and only one control was misclassified.

In the event that 11 instead of 12 cases had been correctly classified, the SN would drop significantly (to 85) and in the event that 14 instead of 15 controls had been correctly classified, the SP would also drop significantly (to 88).

This illustrates that the robustness of the presented classification accuracy statistics is poor as a consequence of the small sample size. The poor validity is also reflected in the wide 95% CIs of the classification accuracy statistics (e.g. an LR+ of 2.2 to 99.1!). In order to calculate robust classification accuracy statistics a sample size of approximately 100 cases and 100 controls is preferable. Smaller sample sizes may be acceptable if the clinical condition examined is rare, but dementia is not a rare condition.

A further problem relates to the concept of ‘memory service-based controls’ (section 2.2.2.2). Patients referred to a memory clinic for assessment should in my opinion not be used as healthy controls in a diagnostic accuracy study like this regardless of the fact that they did not fulfill clinical criteria for a dementia diagnosis. Very little information regarding the cognitive status of the ‘memory service-based controls’ is presented, but it seems likely that some of them may fulfill clinical criteria for mild cognitive impairment.

Reviewer 3 Report

This is a well-written and clearly presented report, and I have only two minor suggestions.

  1. This is a report on the validity of a protocol translated from English to a language developed among Indians emigrated to Fiji Islands. According to Wikipedia, India has 447 different languages related to six language-families. I could be of interest for someone planning a similar project for a possibly related language to know which Indian language and/or language-family  was spoken by the Indians of the Fiji-Indian minority. I suggest that a short note on this should be included in the Introduction-
  2. In Table 2, the acronym MMSE is used. As far as I can see, this acronym has not been explained, and I suggest this is added to the text under the Table.

Round 2

Reviewer 1 Report

.

Author Response

Thank you

Reviewer 2 Report

This reviewer does not agree with the suggestion that an estimate of the diagnostic accuracy of the 10/66 instrument based on 13 cases and 16 controls is better than no information about validated dementia tools in the Fijian-Indian community. I recommend that the researchers continue with the proposed study regarding testing the instrument in the Fiji-Islands, where the Fijian-Indian population is higher and a larger sample of dementia cases and controls can be recruited. The data from the present and the planned study can be pooled and presented in a future manuscript.

Author Response

Thank you for your comments. Fiji and New Zealand are two separate countries and our research team is based in New Zealand. The aim of our project was to assess the 10/66 dementia protocol in Fijian-Indians living in New Zealand (diaspora status being important), not in Fiji. In the first response letter to the reviewer, we explained that the research was conducted in the most practical way possible given that Fijian-Indians aged 65 and over make up less than 1% of the New Zealand population. We believe we have addressed the reviewers’ comments and fully acknowledged that the sample size was small. However, we argued that this was the first such study conducted in the New Zealand Fijian-Indian population and therefore contributes to the research literature. 
We think we need to put the sample size issue in perspective. We have now performed a sample size calculation using the method described by Bujang et al 2016 (1). In our sample size calculation, we used a prevalence of dementia of 60% based on our previously published data from the memory service (2). We found a minimum sample size of 52 subjects (including 31 subjects having dementia) were required to achieve a minimum power of 80%  for detecting a change in the percentage value of sensitivity of a screening test from 0.70 to 0.90, based on a target significance level of 0.05. The methods for this study were described thoroughly in a protocol paper recently accepted by BMJ Open (3), in which we stated our intention to recruit 30 dementia cases and 30 controls.  However, we struggled with participant recruitment. As shown in Figure 1 of our manuscript, a total of 62 Fijian-Indian were initially recruited but 33 of them were subsequently declined or excluded, including a number of them  being reluctant to be interviewed during the COVID-19 pandemic. In the end we were only able to complete interviews with 13 dementia cases and 16 controls.  However, this sample size was still sufficient to demonstrate the 10/66 dementia protocol was valid for use in this population, albeit with wide confidence intervals, and we believe it is still a significant and important finding for the Fijian population living with dementia in New Zealand. The most recent changes (regarding sample size calculation) have been highlighted in yellow in the main manuscript.

  1. Bujang MA, Adnan TH. Requirements for Minimum Sample Size for Sensitivity and Specificity Analysis. J Clin Diagn Res. 2016;10(10)
  2. Cullum S, Mullin K, Zeng I, et al. Do community-dwelling Māori and Pacific peoples present with dementia at a younger age and at a later stage compared with NZ Europeans?. Int J Geriatr Psychiatry. 2018;33(8):1098-1104.
  3. Martinez-Ruiz A, Yates S, Cheung G, et al. Living with Dementia in Aotearoa (LiDiA): A cross-sectional feasibility study protocol for a multi-ethnic dementia prevalence study in Aotearoa/New Zealand. BMJ Open. 2021. In Press